# Lattice-Based Verifiably Encrypted Signature Scheme without Gaussian Sampling for Privacy Protection in Blockchain

**Xiuhua Lu** [1] , **Wei Yin** [2] **and Pingyuan Zhang** [3,]*

1  School of Cyber Science and Engineering, Qufu Normal University, Qufu 273165, China
2  National Computer Network Emergency Response Technical Team/Coordination Center of China (CNCERT/CC), Beijing 100029, China
3  School of Computer and Communication Engineering, Zhengzhou University of Light Industry, Zhengzhou 450002, China
*  Correspondence: jyzhang@zzuli.edu.cn

**Abstract:** Before the transaction data in the blockchain is successfully linked, its signature must be publicly verified by a large number of nodes in the blockchain, which is also one of the ways to leak transaction information. To alleviate the contradiction between the public verifiability of signatures and the protection of transaction privacy, we introduce a verifiably encrypted signature scheme into the blockchain. A verifiably encrypted signature scheme contains two parts of signature information: the encrypted signature is used for public verification, and the ordinary signature is used for internal verification. To reach this goal even better, we design a new lattice-based verifiably encrypted signature scheme, which separates the parameter settings of the signer and the adjudicator, and replaces the Gaussian sampling algorithm with a small range of uniform random sampling, achieving better efficiency and security.

**Keywords:** lattice-based cryptography; verifiably encrypted signature; Gaussian sampling; privacy protection in blockchain

## 1. Introduction

### 1.1. Verifiably Encrypted Signature

A verifiably encrypted signature was first given by Asokan et al. [1] in 1997, which is used to ensure the fairness of the exchange process in a distributed network [2]. Compared with the ordinary signature, the verifiably encrypted signature has an adjudicator besides the signer and verifier. The signer encrypts the ordinary signature using the adjudicator's public key, and the verifier uses the public keys of the signer and the adjudicator to verify the authenticity of the signature ciphertext. If there is a dispute, the adjudicator recovers the signer's ordinary signature from the signature ciphertext. A verifiably encrypted signature is the core of fair contract signing protocols. When the party signing the online contract repudiates, the adjudicator can take the extracted signature as evidence of the signer's signing behavior. In addition to the above applications, a verifiably encrypted signature has many important applications in other fields. Jae Hong Seo et al. [3] have implemented the accumulable optimistic fair exchange using a verifiably encrypted signature. Yujue Wang et al. [4] have introduced a cascading instantiable blank signature on the basis of a verifiably encrypted signature, which realizes the protection of progressive decision management. Therefore, the research on a verifiably encrypted signature has great practical impetus.

A verifiably encrypted signature has achieved good results under the assumption of traditional number theory, such as [5–7]. In pace with the rapid growth of quantum algorithms, verifiably encrypted signatures that can strive against quantum algorithm attacks become more pressing. As the most powerful branch of post-quantum cryptography, lattice-based cryptography has a good degree of performance in the construction of various

cryptographic primitives. A lattice-based verifiably encrypted signature includes [8–10]. To successfully complete the adjudication function of the adjudicator, these schemes have one thing in common: the signer's key depends on the adjudicator's public key. Then, the signer's key will change with the change of the adjudicator, which forms a restriction for the signer to choose the adjudicator. It is of great significance to remove the binding relationship between the key of the signer and the adjudicator in a lattice-based verifiably encrypted signature scheme.

Beyond the above point, as a basic algorithm in lattice-based cryptography [11,12], the Gaussian sampling algorithm has more computational complexity [13] and is vulnerable to side-channel attacks [14,15]. Thomas Prest [13] mentions the following facts. The existing algorithms cannot use discrete Gaussian distribution directly; they have to sample from a statistically approximate distribution. It is generally required that the statistical distance between the sampling distribution and the expected discrete Gaussian distribution is less than $2^{-100}$. To achieve it, a floating-point operation with a precision of at least 100 bits is required. Any precomputation means storing the variable values with the same precision. This may seriously affect the sampling performance on the computer, or even make it impractical to implement on a limited device. With regard to the security, Léo Ducas et al. [16] emphasize the potential side-channel attack risk of Gaussian sampling and suggest replacing Gaussian sampling with random sampling. As for small-range random sampling, the sampling rejection algorithm ensures that the algorithm output will not disclose the signature private key by filtering the output value.

In the wake of the post-quantum cryptography standard collection activities, the efficiency and security of lattice-based cryptosystems have attracted many researchers' attention, and more and more work has been performed to implement the lattice-based cryptosystems, which promotes lattice-based cryptography from the theoretical stage to the practical stage. So far, it is an important research direction to design a secure lattice-based verifiably encrypted signature scheme with better efficiency.

*1.2. Application*

Blockchain is a research field that many cryptographers have recently been paying close attention to, and it is widely used in financial payments such as Bitcoin. At present, digital signature technology is used in the authentication link of blockchain. Specifically, when payer Alice uses Bitcoin or other digital currency to pay payee Bob, she needs to sign the transaction content and broadcast it throughout the whole network, to verify the transaction in the network without an authority center.

Blockchain is an open network, and it is an important demand for privacy protection for traders to complete the public verification of transactions without disclosing the sensitive information of transactions. Transaction information can appear on the blockchain network in the form of a hash digest, but corresponding digital signatures can still disclose sensitive information. Gustavus J. Simmons [17] tell us that sensitive information can be embedded in the random value of the digital signature and transmitted with the message signature pair, which leads to a situation: in the process of signing a transaction, the digital signature may involve sensitive information, and Alice does not want anyone other than Bob to obtain the signature. Some people may think that this is simple: encrypting Alice's signature with Bob's public key and sending the result to Bob. However, we say that this idea is naive: because if the public key encryption scheme is adopted, other participants in the network will no longer be able to publicly verify transactions and signatures, which goes against the original intention of the decentralized blockchain.

We introduce verifiably encrypted signatures into the blockchain, which will solve the problem that signatures need to be publicly verified and that signatures need to avoid the disclosure of transaction information. The verifiably encrypted signature scheme designed in this paper can reduce privacy leakage in the blockchain. In addition, our scheme can also be used in other environments with privacy protection requirements for information authentication, such as image encryption authentication privacy protection [18,19].

### *1.3. Our Contribution*

1. Taking the verifiably encrypted signature in a lattice as the research object, we investigate the limitations caused by the correlation between the parameters of the signer and the adjudicator, the important role of the Gaussian sampling algorithm in lattice signatures, and the fact that the Gaussian sampling algorithm is vulnerable to side-channel attacks. On this basis, a new and verifiably encrypted signature scheme based on the assumption of the lattice difficulty problem is designed. The new scheme realizes the relative independence of the keys of the signer and the adjudicator and avoids the pre-set communication between the two parties. We replace the Gaussian sampling algorithm with a small range of uniform random sampling, which enhances the scheme's security and efficiency.

2. We analyze the double needs of blockchain for the authentication and privacy of signatures and interpret verifiably encrypted signatures in a manner suitable for the blockchain environment. We regard the transaction's initiator as the signer, the transaction's receiver as the adjudicator, and the verifier who can verify the encrypted signature as other public nodes in the blockchain. We embed verifiably encrypted signatures into the blockchain environment and realize the public authentication and privacy protection of transactions with the public verification and arbitration verification functions of verifiably encrypted signatures.

### *1.4. Paper Outline*

The subsequent content of this paper includes the following six aspects. In Section 2, we display some notations and facts, as well as the module short integer solution problem and the computational ring-LWR assumption. In Section 3, we describe the verifiably encrypted signature's definition and security model, and the basic structure of the blockchain. In Section 4, we design the verifiably encrypted signature scheme without Gaussian sampling from a lattice, which is suitable for blockchain scenarios. Our scheme's correctness analysis is also here. In Section 5, we analyze our scheme's security, including strong unforgeability, strong opacity, and extractability. In Section 6, we make a comparison between the previous related schemes and point out the application mode and the special role of our scheme in the blockchain. Lastly, a conclusion is given in Section 7.

### 2. Preliminaries

#### *2.1. Notations*

The symbols in the paper mainly come from [16,20].

$R = \mathbb{Z}[X]/(X^n + 1)$ and $R_q = \mathbb{Z}_q[X]/(X^n + 1)$ are two polynomial rings.

For integer $w \in \mathbb{Z}_q$, $\|w\|_\infty = |w \bmod q|$. For $w = w_0 + w_1 X + \cdots + w_{n-1} X^{n-1} \in R_q$, $\|w\|_\infty = max_i \|w_i\|_\infty$. For $\mathbf{w} = (w_1, \cdots, w_k) \in R_q^k$, $\|\mathbf{w}\|_\infty = max_i \|w_i\|_\infty$.

$U_{\hat{\beta}}$ denotes uniform distribution in $[-\hat{\beta}, \hat{\beta}]$, $U_{\hat{\beta}}^n = \{w = \sum_{i=0}^{n-1} w_i X^i \in R_{\hat{q}} | w_i \leftarrow U_{\hat{\beta}}, i = 0, \cdots, n-1\}$, $(U_{\hat{\beta}}^n)^\times = \{w = \sum_{i=0}^{n-1} w_i X^i \in R_{\hat{q}}$ is invertible $| w_i \leftarrow U_{\hat{\beta}}, i = 0, \cdots, n-1\}$.

For $S_\eta = \{w \in R_q \mid \| w \|_\infty \leq \eta\}$, $w$ is a polynomial with coefficients that are less than or equal to $\eta$ in $R_q$, and $S_\eta^l = \{\mathbf{w} \in R_q^l \mid \| w_i \|_\infty \leq \eta, i = 1, 2, \cdots, l\}$.

$B_{60} = \{c \in R_q | The\ coefficients\ of\ c\ have\ 60\ positive\ and\ negative\ ones,\ and\ the\ others\ are\ zeros\}$.

For $2 \leq p \leq \hat{q}$, integer $x$, $\bar{x} = x \bmod \hat{q}$, floor rounding function $\lfloor \cdot \rfloor_p : \mathbb{Z}_{\hat{q}} \to \mathbb{Z}_p$ is defined as: $\lfloor x \rfloor_p = \lfloor (p/\hat{q}) \cdot \bar{x} \rfloor \bmod p$, function $Inv(\cdot) : \mathbb{Z}_p \to \mathbb{Z}_{\hat{q}}$ is defined as: $Inv(x) \leftarrow \{y \in \mathbb{Z}_{\hat{q}} | \lfloor y \rfloor_p = x\}$.

Reconciliation rounding function $[\cdot]_{2,\hat{q}} : x \to \lfloor \frac{2}{\hat{q}} \cdot x \rfloor \bmod 2$, reconciliation cross-rounding function: $\langle \cdot \rangle_{2,\hat{q}} : x \to \lfloor \frac{4}{\hat{q}} \cdot x \rfloor \bmod 2$.

The algorithm rec, with input $y \in \mathbb{Z}_{\hat{q}}$ and $z \in \{0, 1\}$, output $[x]_{2,\hat{q}}$, where $x$ is the element with the smallest distance from $y$, such as $\langle x \rangle_{2,\hat{q}} = z$.

The randomized doubling function $dbl: \mathbb{Z}_{\hat{q}} \to \mathbb{Z}_{2\hat{q}}, x \mapsto 2x - e$, where $e$ samples from $\{-1, 0, 1\}$ with probabilities $p_{-1} = p_1 = 1/4$, $p_0 = 1/2$.

Functions $f(n), g(n): \mathbb{N} \to \mathbb{R}^+$, $f(n) = \Omega(g(n))$ denote that there exist two constants $U, V$ such that $g(n) \leq U \cdot f(n)$ for all $n \geq V$.

At the end of this subsection, we summarize the basic symbols used in the text into Table 1 for easy searching.

**Table 1.** Symbol Description.

| Symbols | Symbolic Meaning |
|---|---|
| $\mathbb{N}$ | natural numbers set |
| $\mathbb{Z}$ | integers set |
| $\mathbb{R}$ | real numbers set |
| $\mathbb{R}^+$ | positive real numbers set |
| $\lfloor x \rfloor$ | the largest integer not exceeding $x$ |
| $\lceil x \rceil$ | the smallest integer not less than $x$ |
| $Bit(x)$ | binary representation of $x$ |
| $x \leftarrow S$ | $x$ is uniform random in set $S$ |
| $R_q$ | polynomial rings |
| $\|w\|_\infty$ | $max_i \|w_i\|_\infty$ |
| $U_{\hat{\beta}}$ | uniform distribution in $[-\hat{\beta}, \hat{\beta}]$ |
| $S_\eta$ | the set of polynomials with coefficients less than or equal to $\eta$ in $R_q$ |
| $B_{60}$ | $\{c \in R_q | The\ coefficients\ of\ c\ have\ 60\ positive\ and\ negative\ ones,\ and\ the\ others\ are\ zeros\}$ |
| $\bar{x}$ | $x \bmod \hat{q}$ |
| $\lfloor x \rfloor_p$ | $\lfloor (p/\hat{q}) \cdot \bar{x} \rfloor \bmod p$ |
| $Inv(x)$ | $Inv(x) \leftarrow \{y \in \mathbb{Z}_{\hat{q}} | \lfloor y \rfloor_p = x\}$ |
| $[x]_{2,\hat{q}}$ | $\lfloor \frac{2}{\hat{q}} \cdot x \rfloor \bmod 2$ |
| $\langle x \rangle_{2,\hat{q}}$ | $\lfloor \frac{4}{\hat{q}} \cdot x \rfloor \bmod 2$ |
| $dbl(x)$ | $2x - e$ |

### 2.2. Lattice Problems and Facts

**Definition 1** ([21]). *$M - SIS_{q,m,\beta}$ is defined as follows: Given $\mathbf{a}_1, \mathbf{a}_2, \cdots, \mathbf{a}_m \in R_q^d$, which are uniform and independent, find $z_1, z_2, \cdots, z_m \in R$, such that $\Sigma_{i=1}^m \mathbf{a}_i z_i = 0 \bmod q$ and $0 < \|\mathbf{z}\| \leq \beta$, where $\mathbf{z} = (z_1, z_2, \cdots, z_m)^T \in R^m$.*

The module short integer solution problem (M-SIS) is a generalization of the short integer solution problem (SIS) and the ring short integer solution problem (R-SIS), whose hardness is based on the module shortest independent vectors problem (Mod-SIVP).

**Definition 2** ([20]). *$s$ is selected from a distribution $\chi$ over $R$. Let $\chi_s$ be the distribution of $(a, \lfloor as \rfloor_p)$, where $a \leftarrow R_{\hat{q}}$, and let $\mathcal{U}$ be the distribution of $(a, \lfloor b \rfloor_p)$, where $a, b \leftarrow R_{\hat{q}}$. Denote $\mathcal{S}_1 = (\chi_s^l, \mathcal{D})$ and $\mathcal{S}_2 = (\mathcal{U}^l, \mathcal{D})$, $\mathcal{D} = \{0, 1\}^*$. For a challenger $\mathcal{C}$, $P_{\mathcal{C}, \mathcal{A}}(\chi)$ is the probability for an adversary $\mathcal{A}$ to win $Exp_1(\mathcal{C}, \mathcal{A})$ with $\mathcal{S}_1$; $Q_{\mathcal{C}, \mathcal{A}}(\chi)$ is the probability for an adversary $\mathcal{A}$ to win $Exp_2(\mathcal{C}, \mathcal{A})$ with $\mathcal{S}_2$.*

*The computational ring-LWR assumption with respect to a secret distribution $\chi$ says that for all challengers $\mathcal{C}$, if $Q_{\mathcal{C}, \mathcal{A}}$ is negligible for any adversary $\mathcal{A}$, $P_{\mathcal{C}, \mathcal{A}}$ does so.*

The computational ring-LWR assumption with respect to a secret distribution $\chi$, also as $R - CLWR_\chi$, is based on the approximate shortest independent vectors problem (app-SIVP).

**Lemma 1** ([20]). *If $\hat{q}$ is odd and $|x - y| < \hat{q}/8$, then $rec(y, \langle dbl(x) \rangle_{2,2\hat{q}}) = [dbl(x)]_{2,2\hat{q}}$.*

**Lemma 2** ([22]). *$B_1 = B_1(\lambda)$ and $B_2 = B_2(\lambda)$ are two positive integers, $e_1 \in [-B_1, B_1]$ is a fixed integer, and $e_2 \leftarrow [-B_2, B_2]$. If $\frac{B_1}{B_2}$ is negligible, then the statistical distance between the distribution of $e_2$ and the distribution of $e_2 + e_1$ is also negligible.*

### 3. General Model of the Verifiably Encrypted Signature and Blockchain

*3.1. Definition of Verifiably Encrypted Signature*

For the verifiably encrypted signature's definition and security model, we refer to Kee Sung Kim and Ik Rae Jeong [10]. A verifiably encrypted signature scheme involves three parties: the signer, verifier, and adjudicator. The signer is responsible for generating the ordinary signature and the verifiably encrypted signature of the message, the verifier is responsible for the verification of two kinds of signatures, and the adjudicator is responsible for the ordinary signature extraction of the verifiably encrypted signature to prevent the signer's malicious repudiation. The three parties work together to complete the following algorithms.

- **Setup** ($\lambda$): $\lambda$ is the security parameter as input; this algorithm outputs $PP$ as the system public parameter.
- **AKeyGen** ($\lambda$): The adjudicator provides public key $apk$ and secret key $ask$, which are used to generate the signature ciphertext and extract an ordinary signature.
- **KeyGen** ($\lambda$): The signer provides a secret signing key $sk$ and a public verification key $vk$, which are used to generate and verify the ordinary signature, respectively.
- **Sign** ($sk, M$): With the signing key $sk$, the signer provides an ordinary signature $\sigma$ for message $M$.
- **Verify** ($vk, (M, \sigma)$): Given message $M$ and its signature $\sigma$ associated with the verification key $vk$, the verifier and the adjudicator determine whether the ordinary signature $\sigma$ provides legal authentication for the message $M$. If the answer is yes, they output 1, indicating approval of the authentication; otherwise, they output 0, indicating a denial of the authentication.
- **VES-Sign** ($sk, M, apk$): With signing key $sk$, message $M$ and the adjudicator's public key $apk$, the signer provides the verifiably encrypted signature $\delta$.
- **VES-Verify** ($vk, apk, (M, \delta)$): Given the signer's verification key $vk$, the adjudicator's public key $apk$, and message $M$, as well as its verifiably encrypted signature $\delta$, the verifier and the adjudicator determine whether the signature $\delta$ provides legal authentication for the message $M$. If the answer is yes, they output 1, indicating approval of the authentication; otherwise, they output 0, indicating denial of the authentication.
- **Adju** ($ask, vk, (M, \delta)$): With the adjudicator's secret key $ask$, the signer's verification key $vk$ and message $M$, as well as its verifiably encrypted signature $\delta$, the adjudicator extracts an ordinary signature $\sigma$ from $\delta$ for message $M$.

The correctness of the scheme includes two aspects.

- $(M, \delta)$, the output of algorithm **VES-Sign** ($sk, M, apk$), needs to be verified by algorithm **VES-Verify** ($vk, apk, (M, \delta)$).
- $(M, \sigma)$, the output of algorithm **Adju** ($ask, vk, (M, \delta)$), needs to be verified using algorithm **Verify** ($vk, (M, \sigma)$).

*3.2. Security Model of the Verifiably Encrypted Signature*

A verifiably encrypted signature scheme should satisfy strong unforgeability, strong opacity, extractability, and abuse-freeness. Because strong unforgeability implies abuse-freeness, we only consider three security definitions.

3.2.1. Strong Unforgeability

- **Initialization**: Challenger $\mathcal{C}$ executes the algorithms **Setup**, **AKeyGen**, and **KeyGen**, and obtains the public parameters $PP$, the adjudicator's secret key $ask$, and the public key $apk$, as well as the signer's signing key $sk$ and verification key $vk$. Then, challenger $\mathcal{C}$ provides the public parameters $PP$, the adjudicator's secret key $ask$ and public key $apk$, and the signer's verification key $vk$ to adversary $\mathcal{A}$.
- **Verifiably Encrypted Signature Queries**: Adversary $\mathcal{A}$ adaptively performs verifiably encrypted signature queries with a polynomial bound.

Adversary $\mathcal{A}$ selects message $M$ and sends it to challenger $\mathcal{C}$ for the associated verifiably encrypted signature. Challenger $\mathcal{C}$ invokes the **VES-Sign** algorithm, and returns the result to adversary $\mathcal{A}$. Adversary $\mathcal{A}$ can adaptively execute the query polynomial.

- **Forgery**: When adversary $\mathcal{A}$ finishes the queries, he gives a fresh message $M^*$ and its verifiably encrypted signature $\delta^*$.

If a message–signature pair $(M^*, \delta^*)$ can pass the **VES-Verify** algorithm, and it is not the result of some verifiably encrypted signature query, adversary $\mathcal{A}$ wins the game.

**Theorem 1.** *A verifiably encrypted signature scheme owns strong unforgeability, if, for every adversary $\mathcal{A}$ with polynomial bounded computational resources, the probability of him winning the above game is negligible.*

3.2.2. Strong Opacity

- **Initialization**: Challenger $\mathcal{C}$ executes algorithms **Setup**, **AKeyGen**, and **KeyGen**, obtains public parameters $PP$, and the adjudicator's secret key $ask$ and public key $apk$, as well as the signer's signing key $sk$ and verification key $vk$. Then, challenger $\mathcal{C}$ provides public parameters $PP$, the adjudicator's public key $apk$, and the signer's verification key $vk$ to adversary $\mathcal{A}$.
- **Queries**: Adversary $\mathcal{A}$ adaptively performs verifiably encrypted signature queries and adjudication queries with a polynomial bound.

  1. **VES-Sign Query**: Adversary $\mathcal{A}$ selects message $M$, and sends it to challenger $\mathcal{C}$ for the associated verifiably encrypted signature. Challenger $\mathcal{C}$ invokes the **VES-Sign** algorithm and returns the result to adversary $\mathcal{A}$. Adversary $\mathcal{A}$ can adaptively execute the query polynomial.
  2. **AdjuQuery**: Adversary $\mathcal{A}$ sends $(M, \delta)$ to challenger $\mathcal{C}$ for the associated ordinary signature. Challenger $\mathcal{C}$ invokes the **Adju** algorithm, and returns the result to adversary $\mathcal{A}$. Adversary $\mathcal{A}$ can adaptively execute the query polynomial.

- **Forgery**: When adversary $\mathcal{A}$ finishes the queries, he gives a fresh message $M^*$ and its ordinary signature $\sigma^*$.

If a message–signature pair $(M^*, \sigma^*)$ can pass the **Verify** algorithm and it is not the result of some adjudication query, adversary $\mathcal{A}$ wins the game.

**Theorem 2.** *A verifiably encrypted signature scheme owns strong opacity if, for every adversary $\mathcal{A}$ with polynomial bounded computational resources, the probability of him winning the above game is negligible.*

3.2.3. Extractability

- **Initialization**: Challenger $\mathcal{C}$ executes the algorithms **Setup** and **AKeyGen**, and obtains public parameters $PP$, the adjudicator's secret key $ask$, and public key $apk$. Then, challenger $\mathcal{C}$ provides the public parameters $PP$ and the adjudicator's public key $apk$ to adversary $\mathcal{A}$.
- **AdjuQueries**: Adversary $\mathcal{A}$ adaptively performs adjudication queries with a polynomial bound.
  Adversary $\mathcal{A}$ sends $(sk, vk, M, \delta)$ to challenger $\mathcal{C}$ for the associated ordinary signature. Challenger $\mathcal{C}$ invokes the **Adju** algorithm and returns the result to adversary $\mathcal{A}$. Adversary $\mathcal{A}$ can adaptively execute the query polynomial.
- **Forgery**: When adversary $\mathcal{A}$ finishes the queries, he gives a message $M^*$ and its verifiably encrypted signature $\delta^*$, as well as a signer's key pair $(sk^*, vk^*)$.

If the message–signature pair $(M^*, \delta^*)$ can pass the **VES-Verify** algorithm, and the result of algorithm **Adju** $(ask, vk^*, (M^*, \delta^*))$ is invalid, adversary $\mathcal{A}$ wins the game.

**Theorem 3.** *A verifiably encrypted signature scheme owns extractability, if, for every adversary $\mathcal{A}$ with polynomial bounded computational resources, the probability of him winning the above game is negligible.*

*3.3. The Structure of Blockchain*

The blockchain is a chain composed of a large number of blocks. Blocks are generated via an enormous number of distributed network nodes through a consensus algorithm. Each block records different transaction contents. In a blockchain, each node can be considered a user. Each user has a unique address information identification, the address information comes from the user's public key, and the private key is held locally by the user. When user **A** attempts to initiate a transaction with user **B**, he signs the transaction. The transaction will lock a payout and claim that only recipients who meet the lockup conditions will be the owner of the money. To be specific, user **A** signs the transaction using his private key, claiming that the money can only be spent by providing recipient **B**'s legitimate signature. User **A** marks **B** with user **B**'s address, which can be a string of numbers. Because this condition is met, the only user who can provide **B**'s signature is **B** himself, and so funds are safely transferred from **A** to **B**. Each node in the blockchain performs the following tests after receiving a transaction:

1. Check whether the signature in the transaction is valid or not, and reject it if the signature is invalid.
2. Check whether the delivery address has sufficient funds to complete the transaction, and reject the transaction if the balance is insufficient.
3. Update the blockchain ledger based on the consensus mechanism.

The decentralized design not only reduces the risk of network congestion and collapse, but also protects the privacy and freedom of users. However, for some specific focus events, such as a huge transaction on the blockchain during the time period when a company or an individual needs to pay a huge amount of money, even if there is no identity information of both parties in the transaction, people will associate the possible link between them and obtain the property distribution of the parties through the public transaction information. Identity anonymity is far from enough for real life, where identity can be locked through multiple channels. If transaction information and corresponding signatures can obtain more privacy protection, the blockchain can give people more of a sense of security.

**4. Lattice-Based Verifiably Encrypted Signature Scheme without Gaussian Sampling**

Our scheme is based on Fiat–Shamir style lattice-based signature schemes [23]; we use the construction framework of the scheme [23] and the optimization algorithm of the scheme [16] is also applicable. In our scheme, the signer's key generation algorithm and the adjudicator's key generation algorithm are relatively independent, which avoids the interaction between the signer and the adjudicator in the key generation phase. We use a small range of uniform random sampling algorithms instead of a Gaussian sampling algorithm to reduce the impact of side-channel attacks and the computational complexity of the scheme.

*4.1. Design*

- **Setup** ($n$): The system parameters, sets, and functions involved in the scheme are defined as follows.
    1. $q = 2^{23} - 2^{13} + 1 = 8{,}380{,}417$, $n = 256$, $\eta = 5$, $k = 5$, $l = 4$, $\gamma_1 = (q-1)/16 = 523{,}776$, $\beta = 275$, $\hat{q} = \Omega(n^{6.5} \log n)$ is odd, $p = n^{3.75} \log^{1/4} n$, $\hat{\beta} = n^{2.75} \log^{1/4} n$.
    2. Function $G : \{0,1\}^{256} \longrightarrow R_{\hat{q}}$ is defined as in [20].
    3. Functions $H_1 : \{0,1\}^* \longrightarrow B_{60}$ and $H_2 : \{0,1\}^{256} \to \{0,1\}^{\lceil nl \log_2 3 \rceil}$ are collision-resistant hash functions.

- **AKeyGen** ($n$): The adjudicator selects $\tau \leftarrow \{0,1\}^{256}$, $\hat{s} \leftarrow (U_{\hat{\beta}}^n)^{\times}$ and computes $a = G(\tau)$, $b = \lfloor \hat{s}a \rceil_p$. Then, he provides a public key $apk = (\tau, b)$ and a secret key $ask = \hat{s}$.

- **KeyGen** ($n$): The signer samples $\mathbf{A} \leftarrow R_q^{k \times l}$, $\mathbf{s} \leftarrow S_\eta^l$, and computes $\mathbf{t} = \mathbf{As}$. Then, verification key $vk = (\mathbf{A}, \mathbf{t})$, and signing key $sk = \mathbf{s}$.

- **Sign** ($sk = \mathbf{s}, \mu \in \{0,1\}^*$): The signer obtains the ordinary signature $\sigma$ with respect to the signing key $sk = \mathbf{s}$ and the message $\mu \in \{0,1\}^*$.
    1. Sample $\alpha \leftarrow R_q^k$, $\mathbf{y} \leftarrow S_{\gamma_1-1}^l$, and compute $\mathbf{w} = \mathbf{Ay} + \alpha$, $c = H_1(\mu, \mathbf{w})$, $\mathbf{z} = \mathbf{y} + c\mathbf{s}$.
    2. If $\|\mathbf{z}\|_\infty \geq \gamma_1 - \beta$, repeatedly sample $\mathbf{y} \leftarrow S_{\gamma_1-1}^l$, and compute $\mathbf{w} = \mathbf{Ay} + \alpha$, $c = H_1(\mu, \mathbf{w})$, $\mathbf{z} = \mathbf{y} + c\mathbf{s}$.
    3. If $\|\mathbf{z}\|_\infty < \gamma_1 - \beta$, return $\sigma = (\mathbf{z}, c, \alpha)$ as the signature of message $\mu$.

- **Verify** ($vk = (\mathbf{A}, \mathbf{t}), (\mu, \sigma = (\mathbf{z}, c, \alpha))$): Given message $\mu$ and its signature $\sigma = (\mathbf{z}, c, \alpha)$ associated with verification key $vk = (\mathbf{A}, \mathbf{t})$, the verifier make the following judgment. If $c = H_1(\mu, \mathbf{Az} - c\mathbf{t} + \alpha)$ and $\|\mathbf{z}\|_\infty < \gamma_1 - \beta$ holds, the signature is valid and output 1; otherwise, the signature is invalid and output 0.

- **VES-Sign** ($sk = \mathbf{s}, \mu, apk = (\tau, b)$): With signing key $sk = \mathbf{s}$, message $\mu$, and the adjudicator's public key $apk = (\tau, b)$, the signer computes the verifiably encrypted signature $\delta$.
    1. Sample $\mathbf{y}_1 \leftarrow S_{\gamma_1-1}^l$, $\mathbf{y}_2 \leftarrow S_1^l$, and compute $\mathbf{w} = \mathbf{Ay}_1 + \mathbf{Ay}_2$, $c = H_1(\mu, \mathbf{w})$, $\mathbf{z}_1 = \mathbf{y}_1 + c\mathbf{s}$, $\mathbf{z} = \mathbf{y}_1 + c\mathbf{s} + \mathbf{y}_2$.
    2. If $\|\mathbf{z}_1\|_\infty \geq \gamma_1 - \beta$ or $\|\mathbf{z}\|_\infty \geq \gamma_1 - \beta$, repeatedly sample $\mathbf{y}_1 \leftarrow S_{\gamma_1-1}^l$, and compute $\mathbf{w} = \mathbf{Ay}_1 + \mathbf{Ay}_2$, $c = H_1(\mu, \mathbf{w})$, $\mathbf{z}_1 = \mathbf{y}_1 + c\mathbf{s}$, $\mathbf{z} = \mathbf{y}_1 + c\mathbf{s} + \mathbf{y}_2$. The operations end when $\|\mathbf{z}_1\|_\infty < \gamma_1 - \beta$ and $\|\mathbf{z}\|_\infty < \gamma_1 - \beta$.
    3. Compute $\alpha = \mathbf{Ay}_2$ and construct the corresponding non-interactive zero-knowledge proof $\pi$ due to [24].
    4. Sample $r \leftarrow (U_{\hat{\beta}}^n)^{\times}$, and let $\bar{v} = \lfloor Inv(b)r \rceil_p$, $\hat{v} = Inv(\bar{v})$, $v = \langle dbl(\hat{v}) \rangle_{2,2\hat{q}}$.
    5. Let $a = G(\tau)$, $u = \lfloor ra \rceil_p$, $\varpi = H_2([dbl(\hat{v})]_{2,2\hat{q}}) \oplus Bit(\mathbf{y}_2)$.

  Then, verifiably encrypted signature $\delta = (\mathbf{z}, c, \alpha, \pi, u, v, \varpi)$.

- **VES-Verify** ($vk = (\mathbf{A}, \mathbf{t}), apk = (\tau, b), (\mu, \delta = (\mathbf{z}, c, \alpha, \pi, u, v, \varpi))$): Given $vk = (\mathbf{A}, \mathbf{t})$, $apk = (\tau, b)$, and message $\mu$, as well as its verifiably encrypted signature $\delta = (\mathbf{z}, c, \alpha, \pi, u, v, \varpi)$, the verifier makes the following judgment.
    1. Judge the legality of $\pi$. If the result is no, output 0 and reject the signature; otherwise, continue.
    2. If $c = H_1(\mu, \mathbf{Az} - c\mathbf{t})$ and $\|\mathbf{z}\|_\infty < \gamma_1 - \beta$ holds, the signature is valid and output 1; otherwise, the signature is invalid and output 0.

- **Adju** ($ask = \hat{s}, vk = (\mathbf{A}, \mathbf{t}), (M, \delta = (\mathbf{z}, c, \alpha, \pi, u, v, \varpi))$): With $ask = \hat{s}$ $vk = (\mathbf{A}, \mathbf{t})$, and message $M$, as well as its verifiably encrypted signature $\delta = (\mathbf{z}, c, \alpha, \pi, u, v, \varpi)$, the adjudicator extracts an ordinary signature $\sigma$ from $\delta$ for message $M$.
    1. Compute $v' = \hat{s}Inv(u)$ and $\mathbf{y}_2 = Bit^{-1}(\varpi \oplus H_2(rec(v', v)))$.
    2. Let $\mathbf{z}_1 = \mathbf{z} - \mathbf{y}_2$.

  Then, output the ordinary signature $\sigma = (\mathbf{z}_1, c, \alpha)$.

## 4.2. Correctness Analysis

The correctness analysis of the scheme includes the correctness of the ordinary signature, the correctness of the verifiably encrypted signature, and the correctness of the adjudication algorithm. We will elaborate on them separately.

### 4.2.1. The Correctness of the Ordinary Signature

According to the analysis in reference [16], in the ordinary signing algorithm, when the recommended parameters are used, the average number of iterations is

$$e^{n \cdot \beta \cdot l / \gamma_1} = e^{256 \times 275 \times 4 / 523776} \approx 1.71$$

so that the signing algorithm can be effectively terminated, and $\mathbf{z}$ satisfying the condition $\|\mathbf{z}\|_\infty < \gamma_1 - \beta$ can be obtained easily. In addition, because $\mathbf{t} = \mathbf{A}\mathbf{s}$ and $\mathbf{z} = \mathbf{y} + c\mathbf{s}$, we have $\mathbf{A}\mathbf{z} - c\mathbf{t} + \alpha = \mathbf{A}(\mathbf{y} + c\mathbf{s}) - c\mathbf{t} + \alpha = \mathbf{A}\mathbf{y} + c\mathbf{A}\mathbf{s} - c\mathbf{t} + \alpha = \mathbf{A}\mathbf{y} + c\mathbf{t} - c\mathbf{t} + \alpha = \mathbf{A}\mathbf{y} + \alpha = \mathbf{w}$; therefore, $c = H_1(\mu, \mathbf{w}) = H_1(\mu, \mathbf{A}\mathbf{z} - c\mathbf{t} + \alpha)$.

### 4.2.2. The Correctness of the Verifiably Encrypted Signature

In the verifiably encrypted signing algorithm, the probability of $\|\mathbf{z}_1\|_\infty < \gamma_1 - \beta$ is about $e^{-n \cdot \beta \cdot l / \gamma_1}$. When $\|\mathbf{z}_1\|_\infty < \gamma_1 - \beta$, the probability of $\|\mathbf{z}\|_\infty = \gamma_1 - \beta$ is no more than the probability of $\|\mathbf{z}_1\|_\infty = \gamma_1 - \beta - 1$, so that the probability of both $\|\mathbf{z}_1\|_\infty < \gamma_1 - \beta$ and $\|\mathbf{z}\|_\infty < \gamma_1 - \beta$ is about $e^{-n \cdot \beta \cdot l / \gamma_1} \cdot \frac{\gamma_1 - \beta - 1}{\gamma_1 - \beta}$. Therefore, in the verifiably encrypted signing algorithm, when the recommended parameters are used, the average number of iterations is $e^{n \cdot \beta \cdot l / \gamma_1} \cdot \frac{\gamma_1 - \beta}{\gamma_1 - \beta - 1} = e^{256 \times 275 \times 4 / 523776} \times \frac{523776 - 275}{523776 - 275 - 1} \approx 1.71$, so that the verifiably encrypted signing algorithm can be effectively terminated, and $\mathbf{z}_1, \mathbf{z}$ satisfying the condition $\|\mathbf{z}_1\|_\infty < \gamma_1 - \beta$ and $\|\mathbf{z}\|_\infty < \gamma_1 - \beta$ can be obtained easily. Moreover, because $\mathbf{t} = \mathbf{A}\mathbf{s}$ and $\mathbf{z} = \mathbf{y}_1 + c\mathbf{s} + \mathbf{y}_2$, we have $\mathbf{A}\mathbf{z} - c\mathbf{t} = \mathbf{A}(\mathbf{y}_1 + c\mathbf{s} + \mathbf{y}_2) - c\mathbf{t} = \mathbf{A}\mathbf{y}_1 + c\mathbf{A}\mathbf{s} + \mathbf{A}\mathbf{y}_2 - c\mathbf{t} = \mathbf{A}\mathbf{y}_1 + c\mathbf{t} + \mathbf{A}\mathbf{y}_2 - c\mathbf{t} = \mathbf{A}\mathbf{y}_1 + \mathbf{A}\mathbf{y}_2 = \mathbf{w}$; therefore, $c = H_1(\mu, \mathbf{w}) = H_1(\mu, \mathbf{A}\mathbf{z} - c\mathbf{t})$.

### 4.2.3. The Correctness of the Adjudication Algorithm

When $\hat{q} = n^{6.5} \log n$, $p = n^{3.75} \log^{1/4} n$, $\hat{\beta} = n^{2.75} \log^{1/4} n$, $\hat{v} = Inv(\bar{v}) = Inv(b)r + e_1 = (a\hat{s} + e_2)r + e_1 = a\hat{s}r + (e_2 r + e_1)$, $v' = \hat{s}Inv(u) = (ar + e_3)\hat{s} = a\hat{s}r + \hat{s}e_3$, $\hat{v} - v' = (e_2 r + e_1) - \hat{s}e_3$. Due to $\hat{s}, r \leftarrow (U^n_{\hat{\beta}})^\times$, $\|\hat{s}\|_\infty \leq \hat{\beta}$, $\|r\|_\infty \leq \hat{\beta}$, and $|e_1| \leq \hat{q}/p$, $|e_2| \leq \hat{q}/p$, $|e_3| \leq \hat{q}/p$, we have $|e_2 r + e_1| \leq n\hat{\beta}\hat{q}/p + \hat{q}/p$ and $|\hat{s}e_3| \leq n\hat{\beta}\hat{q}/p$, so that $|\hat{v} - v'| = |(e_2 r + e_1) - \hat{s}e_3| \leq 2n\hat{\beta}\hat{q}/p + \hat{q}/p < \hat{q}/8$ with overwhelming probability.

According to Lemma 1, $rec(v', v) = rec(v', \langle dbl(\hat{v}) \rangle_{2,2\hat{q}}) = [dbl(\hat{v})]_{2,2\hat{q}}$. Due to $\omega = H_2([dbl(\hat{v})]_{2,2\hat{q}}) \oplus Bit(\mathbf{y}_2)$, $Bit(\mathbf{y}_2) = \omega \oplus H_2([dbl(\hat{v})]_{2,2\hat{q}}) = \omega \oplus H_2(rec(v', v))$; hence, $\mathbf{y}_2 = Bit^{-1}(\omega \oplus H_2(rec(v', v)))$.

According to the verifiably encrypted signing algorithm, $\mathbf{w} = \mathbf{A}\mathbf{y}_1 + \alpha$, $c = H_1(\mu, \mathbf{w})$, $\mathbf{z}_1 = \mathbf{z} - \mathbf{y}_2 = \mathbf{y}_1 + c\mathbf{s}$, and $\|\mathbf{z}_1\|_\infty < \gamma_1 - \beta$. From the analysis of Section 4.2.1, $\sigma = (\mathbf{z}_1, c, \alpha)$ is a valid ordinary signature.

## 5. Security Analysis of Our Scheme

### 5.1. Strong Unforgeability of Our Scheme

**Theorem 4.** *If there exists adversary $\mathcal{A}$ who can attack the strong unforgeability of our scheme with a probability that cannot be ignored, then challenger $\mathcal{C}$ can find a solution to an M-SIS problem instance with a non-negligible probability by using his ability. In other words, because the M-SIS problem is difficult to solve, our scheme is strongly unforgeable.*

**Proof.** Suppose that adversary $\mathcal{A}$ can forge a verifiably encrypted signature with probability $\epsilon > 0$, and the maximum number of times he executes hash queries is $Q_1$; the maximum number of times that he executes the verifiably encrypted signature queries is $Q_2$. By interacting with adversary $\mathcal{A}$, challenger $\mathcal{C}$ can find the non-zero vector $\mathbf{v} \in R^l$ satisfying the condition $\mathbf{A}\mathbf{v} = \mathbf{0} (\bmod q)$ and $\|\mathbf{v}\|_\infty < 2\gamma_1$, with probability $\frac{\epsilon^2}{2(Q_1 + Q_2)}$ for the M-SIS problem instance $\mathbf{A} \in R^{k \times l}_q$.

- **Initialization**: Challenger $\mathcal{C}$ gives system parameters according to the algorithms **Setup**, and provides the adjudicator's public key $apk = (\tau, b)$ and secret key $ask = \hat{s}$ according to algorithms **AKeyGen**. He also samples $\mathbf{s} \leftarrow S^l_\eta$, computes $\mathbf{t} = \mathbf{A}\mathbf{s}$, and

sets the verification key $vk = (\mathbf{A}, \mathbf{t})$ and signing key $sk = \mathbf{s}$. Then, the challenger $\mathcal{C}$ provides the system parameters, the adjudicator's public key $apk = (\tau, b)$ and secret key $ask = \hat{s}$, and the signer's public verification key $vk = (\mathbf{A}, \mathbf{t})$ to adversary $\mathcal{A}$.

- **Queries**: Adversary $\mathcal{A}$ adaptively performs the following queries with a polynomial bound.

  1. $H_1$**Query**: Challenger $\mathcal{C}$ maintains a list $\mathcal{H}_1$ for $H_1$ queries. When adversary $\mathcal{A}$ sends message $\mu_i$ to challenger $\mathcal{C}$, $\mathcal{C}$ samples $c_i \leftarrow B_{60}$, $\mathbf{z}_i \leftarrow S^l_{\gamma_1 - \beta - 1}$. Then, he selects $\mathbf{y}_{2i} \in S^l_1$, such that $\|\mathbf{z}_i - \mathbf{y}_{2i}\|_\infty < \gamma_1 - \beta$. $\mathcal{C}$ computes $\alpha_i = \mathbf{A}\mathbf{y}_{2i}$, lets $c_i = H_1(\mu_i, \mathbf{A}\mathbf{z}_i - c_i\mathbf{t})$, saves $(\mu_i, \mathbf{z}_i, c_i, \mathbf{y}_{2i})$ in list $\mathcal{H}_1$, and returns $c_i$. When adversary $\mathcal{A}$ sends message $\mu_i$ for the $H_1$ query again, $\mathcal{C}$ returns $c_i$ directly.

  2. **VES-Sign Query**: Adversary $\mathcal{A}$ selects message $\mu_i$ and sends it to challenger $\mathcal{C}$ for the associated verifiably encrypted signature. Challenger $\mathcal{C}$ searches list $\mathcal{H}_1$ for $\mu_i$ and constructs the non-interactive zero-knowledge proof $\pi_i$ for $\alpha_i = \mathbf{A}\mathbf{y}_{2i}$. Then, Challenger $\mathcal{C}$ samples $r_i \leftarrow (U^n_{\hat{\beta}})^\times$ and computes $\bar{v}_i = \lfloor Inv(b)r_i \rfloor_p$, $\hat{v}_i = Inv(\bar{v}_i)$, $v_i = \langle dbl(\hat{v}_i)\rangle_{2,2\hat{q}}$, $a = G(\tau)$, $u_i = \lfloor r_i a \rfloor_p$, $\varpi_i = H_2([dbl(\hat{v}_i)]_{2,2\hat{q}}) \oplus Bit(\mathbf{y}_{2i})$. Finally, Challenger $\mathcal{C}$ returns $\delta_i = (\mathbf{z}_i, c_i, \alpha_i, \pi_i, u_i, v_i, \varpi_i)$ to adversary $\mathcal{A}$.
  If $\mu_i$ does not exist in list $\mathcal{H}_1$, challenger $\mathcal{C}$ executes $H_1$ query for message $\mu_i$ firstly.

- **Forgery**: When adversary $\mathcal{A}$ finishes the queries, he gives a new message $\mu^*$ and its verifiably encrypted signature $\delta^* = (\mathbf{z}^*, c^*, \alpha^*, \pi^*, u^*, v^*, \varpi^*)$, which satisfies $c^* = H_1(\mu^*, \mathbf{A}\mathbf{z}^* - c^*\mathbf{t})$ and $\|\mathbf{z}^*\|_\infty < \gamma_1 - \beta$.

Because adversary $\mathcal{A}$ can make at most $Q_1$ hash queries and $Q_2$ verifiably encrypted signature queries, the number of $c_i$ is, at most, $Q_1 + Q_2$. For an undocumented $\mathbf{w} = \mathbf{A}\mathbf{z} - c\mathbf{t}$, adversary $\mathcal{A}$ has only $\frac{1}{3^{256}}$ probability of producing $c$, such that $c = H_1(\mu, \mathbf{w})$. Therefore, $c^*$ comes from $\{c_1, c_2, \cdots, c_{Q_1 + Q_2}\}$ with probability $1 - \frac{1}{3^{256}}$. In addition, adversary $\mathcal{A}$ forges a valid verifiably encrypted signature with probability $\epsilon$, so that $(\mu^*, \mathbf{z}^*, c^*)$ comes from the valid forgery and $c^* \in \{c_1, c_2, \cdots, c_{Q_1 + Q_2}\}$ with probability $\epsilon - \frac{1}{3^{256}}$. Let $c^* = c_j$, it comes from some $H_1$ query or verifiably encrypted signature query.

If $c^* = c_j$ comes from some $H_1$ query, $\mathcal{C}$ interacts with adversary $\mathcal{A}$ to execute $H_1$ queries and verifiably encrypted signature queries again. According to [25], adversary $\mathcal{A}$ generates a new verifiably encrypted signature $\delta' = (\mathbf{z}', c', \alpha', \pi', u', v', \varpi')$ for message $\mu^*$ with probability $(\epsilon - \frac{1}{3^{256}})(\frac{\epsilon - \frac{1}{3^{256}}}{Q_1 + Q_2} - \frac{1}{3^{256}}) \approx \frac{\epsilon^2}{Q_1 + Q_2}$, where $c' \neq c^*$. This is because $\mathbf{A}\mathbf{z}^* - c^*\mathbf{t} = \mathbf{A}\mathbf{z}' - c'\mathbf{t}$, $\mathbf{t} = \mathbf{A}\mathbf{s}$, so that $\mathbf{A}(\mathbf{z}^* - \mathbf{z}' + c'\mathbf{s} - c^*\mathbf{s}) = \mathbf{0}$. Due to $\|\mathbf{z}^*\|_\infty < \gamma_1 - \beta$, $\|\mathbf{z}'\|_\infty < \gamma_1 - \beta$, $\|c^*\mathbf{s}\|_\infty \leq \beta$, $\|c'\mathbf{s}\|_\infty \leq \beta$, so that $\|\mathbf{z}^* - \mathbf{z}' + c'\mathbf{s} - c^*\mathbf{s}\|_\infty < 2\gamma_1$. If $\mathbf{z}^* - \mathbf{z}' + c'\mathbf{s} - c^*\mathbf{s} \neq \mathbf{0}$, $\mathbf{v} = \mathbf{z}^* - \mathbf{z}' + c'\mathbf{s} - c^*\mathbf{s}$ is a solution. If $\mathbf{z}^* - \mathbf{z}' + c'\mathbf{s} - c^*\mathbf{s} = \mathbf{0}$, there exists $\mathbf{s}' \neq \mathbf{s}$ such as $\mathbf{A}\mathbf{s} = \mathbf{A}\mathbf{s}'$ and $\mathbf{z}^* - \mathbf{z}' + c'\mathbf{s}' - c^*\mathbf{s}' \neq \mathbf{0}$ with overwhelming probability, then $\mathbf{v} = \mathbf{z}^* - \mathbf{z}' + c'\mathbf{s}' - c^*\mathbf{s}'$ is a solution. For adversary $\mathcal{A}$, the occurrence probabilities of $\mathbf{s}'$ and $\mathbf{s}$ are equal, so that $\mathbf{v}$ is obtained with a probability of at least $\frac{1}{2}$.

If $c^* = c_j$ comes from some verifiably encrypted signature query, $H_1(\mu^*, \mathbf{A}\mathbf{z}^* - c^*\mathbf{t}) = H_1(\mu_j, \mathbf{A}\mathbf{z}_j - c_j\mathbf{t})$. If $\mu^* \neq \mu_j$ or $\mathbf{A}\mathbf{z}^* - c^*\mathbf{t} \neq \mathbf{A}\mathbf{z}_j - c_j\mathbf{t}$, then adversary $\mathcal{A}$ finds a preimage of $c_j$. Therefore, $\mu^* = \mu_j$ and $\mathbf{A}\mathbf{z}^* - c^*\mathbf{t} = \mathbf{A}\mathbf{z}_j - c_j\mathbf{t}$, then $\mathbf{A}(\mathbf{z}^* - \mathbf{z}_j) = \mathbf{0}$. Due to $\mathbf{z}^* \neq \mathbf{z}_j$, $\mathbf{z}^* - \mathbf{z}_j \neq \mathbf{0}$. Moveover, $\|\mathbf{z}^*\|_\infty < \gamma_1 - \beta$, $\|\mathbf{z}_j\|_\infty < \gamma_1 - \beta$, then $\|\mathbf{z}^* - \mathbf{z}_j\|_\infty < 2(\gamma_1 - \beta) < 2\gamma_1$. Therefore, $\mathbf{v} = \mathbf{z}^* - \mathbf{z}_j$ is a solution.

In short, whether $c^* = c_j$ comes from $H_1$ query or a verifiably encrypted signature query, challenger $\mathcal{C}$ can find a non-zero vector $\mathbf{v} \in R^l$ satisfying $\mathbf{A}\mathbf{v} = \mathbf{0}(\mathrm{mod}q)$ and $\|\mathbf{v}\|_\infty < 2\gamma_1$ with a probability of $\frac{\epsilon^2}{2(Q_1 + Q_2)}$. $\square$

### 5.2. The Strong Opacity of Our Scheme

In our scheme, message $\mu$'s verifiably encrypted signature is $\delta = (\mathbf{z}, c, \alpha = \mathbf{A}\mathbf{y}_2, \pi, u, v, \varpi)$, and the strong opacity of our scheme equals that $\delta = (\mathbf{z}, c, \alpha = \mathbf{A}\mathbf{y}_2, \pi, u, v, \varpi)$ will not divulge information about $\mathbf{y}_2$.

For $\mathbf{z} = \mathbf{z}_1 + \mathbf{y}_2$, according to Lemma 2, $\mathbf{z}$ and $\mathbf{z}_1$ are statistically indistinguishable, and $\mathbf{z}_1$ has nothing to do with $\mathbf{y}_2$, so that $\mathbf{z}$ will not divulge information about $\mathbf{y}_2$.

According to the hardness of the module short integer solution problem, $\alpha = \mathbf{A}\mathbf{y}_2$ will not divulge information about $\mathbf{y}_2$, and then $c = H_1(\mu, \mathbf{A}\mathbf{y}_1 + \alpha)$ also will not divulge information about $\mathbf{y}_2$.

$\pi$ is the zero-knowledge proof of $\alpha = \mathbf{A}\mathbf{y}_2$, so $\pi$ will not divulge information about $\mathbf{y}_2$.

In conclusion, $\delta = (\mathbf{z}, c, \alpha = \mathbf{A}\mathbf{y}_2, \pi, u, v, \varpi)$ will not divulge information about $\mathbf{y}_2$, so that $\mathbf{z}_1 = \mathbf{z} - \mathbf{y}_2$ cannot be obtained merely by $\delta = (\mathbf{z}, c, \alpha = \mathbf{A}\mathbf{y}_2, \pi, u, v, \varpi)$; hence, our scheme owns strong opacity. For a more rigorous description of strong opacity, see Theorem 5.

**Theorem 5.** *If there exists an adversary $\mathcal{A}$ who can break the strong opacity of our scheme with probability $\epsilon$, then challenger $\mathcal{C}$ can construct an algorithm that can solve the M-SIS problem with a probability of at least $\epsilon/q_H$, where $q_H$ is the maximum number of queries to $H_1$.*

**Proof.** Given an instance of M-SIS problem $\mathbf{A} \in R_q^{k \times l}$, challenger $\mathcal{C}$ needs to find a non-zero short $\mathbf{s}$ satisfying $\mathbf{A}\mathbf{s} = \mathbf{0} \mod q$.

- **Initialization**: Challenger $\mathcal{C}$ executes the algorithms **Setup**, **AKeyGen**, and **KeyGen**, and sends the public key $apk = (\tau, b)$, $vk = (\mathbf{A}, \mathbf{t})$, and $PP$ to the the adversary $\mathcal{A}$.
- **Queries**: Allowed queries include $H$ queries, VES-Sign queries and Adju-Queries. When $\mathcal{A}$ finishes the queries, and with probability $\epsilon$ outputs a forged ordinary signature for some message, Challenger $\mathcal{C}$ can solve the M-SIS problem.
  1. $H$ **Query**: $\mathcal{C}$ first examines the list $L$ for this query $\mu$. If it has not existed in the list $L$, $\mathcal{C}$ randomly chooses $c \in B_{60}$, records the corresponding relationship between $\mu$ and $c$ in the table, and sends $c$ to $\mathcal{A}$. If the query $\mu$ has existed in the list $L$, $\mathcal{C}$ returns its corresponding $c$ to $\mathcal{A}$ directly.
  2. **VES-Sign Query**: $\mathcal{A}$ adaptively chooses message $\mu$, and sends it to the challenger $\mathcal{C}$. $\mathcal{C}$ executes the VES-Sign algorithm, and returns $(\mathbf{z}, c, \alpha, \pi, u, v, \varpi)$ to $\mathcal{A}$.
  3. **AdjuQuery**: Assume that $\mathcal{A}$ has queried to $H$ before Adju-Queries. When receiving the Adju-Queries to the verifiably encrypted signature $(\mathbf{z}, c, \alpha, \pi, u, v, \varpi)$, $\mathcal{C}$ returns the ordinary signature $(\mathbf{z} - Bit^{-1}(\varpi \oplus H_2(rec(\hat{s}Inv(u), v))), c, \alpha)$.

Hence, $\mathcal{A}$ finally proposes a valid ordinary signature $(\mathbf{z}_1, c, \alpha)$ with probability $\epsilon$. If $c$ is a response of the VES-Sign query, there exists another signature $(\mathbf{z}_1', c, \alpha)$ for some message $\mu'$, such that

$$H_1(\mu, \mathbf{A}\mathbf{z}_1 - c\mathbf{t} + \alpha) = H_1(\mu', \mathbf{A}\mathbf{z}_1' - c\mathbf{t} + \alpha).$$

Hence, $\mu = \mu'$, and $\mathbf{A}\mathbf{z}_1 - c\mathbf{t} + \alpha = \mathbf{A}\mathbf{z}_1' - c\mathbf{t} + \alpha$. That is, $\mathbf{A}(\mathbf{z}_1 - \mathbf{z}_1') = \mathbf{0} \mod q$. Note that $\mathcal{A}$ successfully forges a new and valid signature; thus, $\mathbf{s} := \mathbf{z}_1 - \mathbf{z}_1' \neq \mathbf{0}$. Because $\|\mathbf{z}_1\|, \|\mathbf{z}_1'\| \leq \sqrt{l \cdot (\gamma_1 - \beta)^2}$, thus there exists a non-zero vector $\mathbf{s}$, such that $\mathbf{A}\mathbf{s} = \mathbf{0} \mod q$, and $\|\mathbf{s}\| \leq 2\sqrt{l \cdot (\gamma_1 - \beta)^2}$. That is, challenger $\mathcal{C}$ solves the M-SIS problem instance with probability $\epsilon/q_H$. If $c$ is not a response of the VES-Sign query, $\mathcal{C}$ may sign the same message again, and the situation is similar. $\square$

### 5.3. Extractability of Our Scheme

For a verifiably encrypted signature $\delta = (\mathbf{z}, c, \alpha, \pi, u, v, \varpi)$ associated with message $\mu$, if $\delta$ is valid, we can extract an ordinary signature $\sigma$. Proof $\pi$ guarantees the existence of the short vector $\mathbf{y}_2$ in $\alpha = \mathbf{A}\mathbf{y}_2$. With the adjudicator's secret key $ask = \hat{s}$, we can compute $v' = \hat{s}Inv(u)$, $\mathbf{y}_2 = Bit^{-1}(\varpi \oplus H_2(rec(v', v)))$; let $\mathbf{z}_1 = \mathbf{z} - \mathbf{y}_2$, then $\sigma = (\mathbf{z}_1, c, \alpha)$ is an ordinary signature for message $\mu$.

## 6. Comparison of Related Work and Our Scheme's Application in the Blockchain

So far, there has been a lot of work on verifiably encrypted signatures. We mainly compare some of the main schemes in terms of application scenarios, key features, difficulties basis, and resistance to quantum attack. Table 2 shows the details of the comparison. Due to space, some abbreviations are used in Table 2, which are explained as follows. "Resistance to Quantum Attack" is abbreviated as RQA, "cascade-instantiable blank signature" is abbre-

viated as CBS, "adjudicator public key binding" is abbreviated as APKB, "inhomogeneous small integer solution problem" is abbreviated as ISIS, and "Module short integer solution problem" is abbreviated as M-SIS.

**Table 2.** Comparison of Related Work.

| Schemes | Application Scenarios | Key Features | Difficulties Basis | RQA |
|---|---|---|---|---|
| [2] | Ethereum | no adjudicator | Strong Diffie-Hellman assumption | × |
| [3] | optimistic fair exchange | homomorphic | composite order Bilinear groups | × |
| [4] | CBS | non-interactive | prime order Bilinear groups | × |
| [5] | Bitcoin escrow protocol | ECDSA-like | discrete logarithm problem | × |
| [6] | Internet exchange | undeniable signature | discrete logarithm problem | × |
| [7] | online contract signing | obfuscator | decisional linear assumption | × |
| [8] | electronic commence | standard model | short integer solution problem | ✓ |
| [9] | Internet exchange | APKB | ISIS | ✓ |
| [10] | nothing | no adjudicator | ISIS | ✓ |
| Ours | Bitcoin transaction | privacy protection | M-SIS | ✓ |

According to the analysis in Table 2, schemes in the literature [8–10] and our scheme are all lattice-based schemes against quantum algorithm attacks. We further analyze the efficiency of these four schemes in Table 3.

Table 3 lists the comparisons of different verifiably encrypted signature schemes in lattices. m and n represent the dimension and rank of the lattice used in the scheme [8–10], respectively, and $k$ and $l$ represent the dimension and rank of the modular lattice in our scheme, respectively. These four parameters play a decisive role in the verification key size, signing key size, and signature size. Our scheme is based on module lattices, and the values of the corresponding parameters $k, l$ are smaller than that of $m, n$ in general lattices. Thus, the scheme in our work has advantages in terms of signature and key sizes. Furthermore, our construction does not require Gaussian sampling, and it is much simpler to implement it securely against side-channel attacks.

**Table 3.** Comparisons of the Schemes in Lattice.

| Schemes | Verification Key Size | Signing Key Size | VES Size | Gaussian Sampling |
|---|---|---|---|---|
| [8] | $mn \log q$ | $m^2 \log q$ | $3m \log q + l$ | ✓ |
| [9] | $mn \log q$ | $m^2 \log q$ | $2m \log q + n + l$ | ✓ |
| [10] | $2mn \log q$ | $2m^2 \log q$ | $(2m + m^2) \log q + n + l$ | ✓ |
| Ours | $k(l+1) \log q$ | $l \log q$ | $(k+l) \log q + n + k$ | × |

When our scheme is applied to the blockchain scenario, there are three participants: the payer Alice (signer), the payee Bob (digital signature receiver), and the verifier (miner in the blockchain). More precisely, these three parties are nodes in the blockchain network. After generating a transaction between Alice and Bob, Alice signs the transaction with her private key, associates Bob's public key with the signature to obtain the verifiably encrypted signature, then broadcasts the result to the blockchain network. The miner in charge of keeping a ledger verifies the signature and records it. Each miner can verify the verifiably encrypted signature to prove the real existence of the transaction, but they cannot obtain more information about the transaction and both parties from the signature. The payee Bob has the private key used for the encrypted signature and is able to obtain the common signature of the transaction for further confirmation of the transaction, and as evidence to avoid disputes with the payer Alice. Figure 1 shows the basic framework.

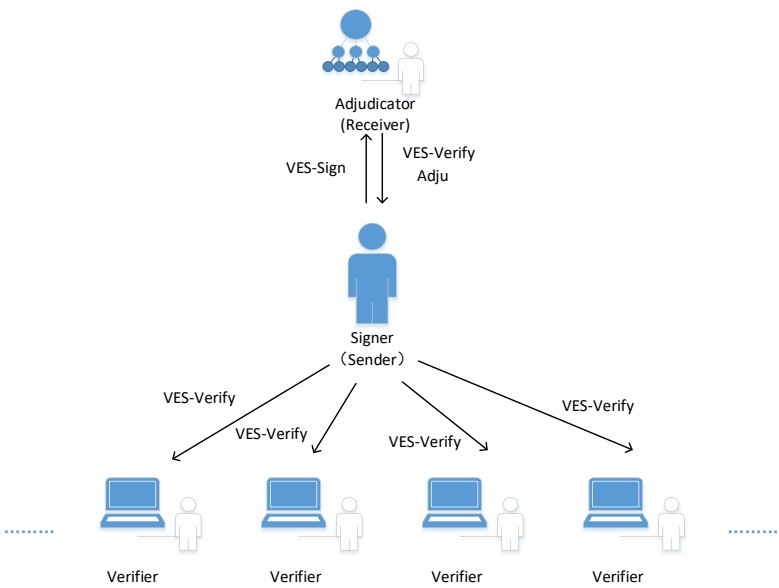

**Figure 1.** The Framework of Our Scheme.

Introducing verifiably encrypted signatures into the blockchain has two functions. First, publicly verifiable signatures are encrypted. Verifiably encrypted signatures prevent blockchain nodes from obtaining transaction information through this signature while ensuring the signature authentication function. Second, for the recipient of the transaction, he still obtains the ordinary signature of the transaction, so that the displayed authentication of the transaction information under his control is realized. A verifiably encrypted signature balances the public verification demand of the signature, the privacy demand of the transaction party, and the controllability of the arbitration demand, to a certain extent.

## 7. Conclusions

We construct a new and verifiably encrypted signature scheme in the lattice; the scheme realizes the relative independence of the signer and adjudicator, eliminates dependence on the Gaussian sampling algorithm, simplifies the parameter setting process of the participants, enhances the security, and improves the operation efficiency, which is a better choice for the actual applications. We integrate this signature scheme into the blockchain environment, which not only realizes the public verification requirements of the blockchain for transactions, but also reduces the disclosure of information about the privacy of transactions from the disclosure of signatures in the blockchain to a certain extent. Our scheme provides a good choice for blockchain transaction authentication. In our environment, the initiator and receiver of a transaction need to consult with each other about the transaction information, which is a natural situation in real life. If message recoverability is added to the signature, this restriction is no longer necessary. The last thing we want to say is that, in the blockchain environment, we have given the signer, verifier, and adjudicator a new role and a new idea for the verifiably encrypted signature scheme's application. We believe that this idea can be extended to more application environments that are sensitive to signature privacy.

**Author Contributions:** Conceptualization, X.L.; methodology, X.L.; validation, X.L., W.Y. and P.Z.; formal analysis, X.L. and P.Z.; writing—original draft preparation, X.L. and W.Y.; writing—review and editing, X.L. and W.Y. All authors have read and agreed to the published version of the manuscript.

**Funding:** This research was funded by the National Cryptography Development Fund, grant number MMJJ20180110; the National Natural Science Foundation of China, grant number 62102300; Shandong Social Science Planning Project, grant number 21CSDJ30; and Key Research Project of Higher Education Institutions of Henan Province, grant number 23A520012.

**Institutional Review Board Statement:** Our study does not involve humans or animals, nor state secrets or confidential projects. The names "Alice" and "Bob" used in our study are two commonly used personas in cryptography. They do not refer to specific characters and do not have infringement attributes.

**Informed Consent Statement:** Not applicable.

**Data Availability Statement:** Not applicable.

**Acknowledgments:** Thank reviewers and the editorial department for their suggestions.

**Conflicts of Interest:** The authors declare no conflict of interest.

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
