# Peer review of "Lattice-Based Verifiably Encrypted Signature Scheme without Gaussian Sampling for Privacy Protection in Blockchain"

_sustainability, doi:10.3390/su142114225_

Round 1

Reviewer 2 Report

The authors proposed a lattice based verifiably encrypted scheme for blockchain. 

Although, there is significant discussion on small range random sampling instead of Guassian sampling, however, there are no practical results as to how this will speed up the process. Simulation results/ discussion should be incorporated.  Moreover, its security should also be discussed, as to how the security is not compromised. 

The comparison in Table 1 with ref 6, 7, 8 seems odd. Comparison with the latest lattice based signatures schemes should be incorporated. The symbols used for comparison in Table 1 should be explicitly defined to compel the readers regarding the efficiency. 

Round 2

Reviewer 2 Report

The authors have tried to address my arguments. 

Reviewer 3 Report

Please do work on sentence structure and modify it on English proofreading professional. Further the paper merits for acceptance.